# Potentiation of Collagen Deposition by the Combination of Substance P with Transforming Growth Factor Beta in Rat Skin Fibroblasts

**DOI:** 10.3390/ijms25031862

**Published:** 2024-02-03

**Authors:** Brendan A. Hilliard, Mamta Amin, Steven N. Popoff, Mary F. Barbe

**Affiliations:** 1Aging and Cardiovascular Discovery Center, Lewis Katz School of Medicine at Temple University, Philadelphia, PA 19140, USA; mamta@temple.edu (M.A.); mary.barbe@temple.edu (M.F.B.); 2Department of Biomedical Education, Lewis Katz School of Medicine at Temple University, Philadelphia, PA 19140, USA; steven.popoff@temple.edu

**Keywords:** substance p, TGFbeta, fibroblasts, collagen, proliferation, smooth muscle actin

## Abstract

A role for substance P has been proposed in musculoskeletal fibrosis, with effects mediated through transforming growth factor beta (TGFβ). We examined the in vitro effects of substance P on proliferation, collagen secretion, and collagen deposition in rat primary dermal fibroblasts cultured in medium containing 10% fetal bovine serum, with or without TGFβ. In six-day cultures, substance P increased cell proliferation at concentrations from 0.0002 to 100 nM. TGFβ increased proliferation at concentrations from 0.0002 to 2 pg/mL, although higher concentrations inhibited proliferation. Substance P treatment alone at concentrations of 100, 0.2, and 0.00002 nM did not increase collagen deposition per cell, yet when combined with TGFβ (5 ng/mL), increased collagen deposition compared to TGFβ treatment alone. Substance P treatment (100 nM) also increased smooth muscle actin (SMA) expression at 72 h of culture at a level similar to 5 ng/mL of TGFβ; only TGFβ increased SMA at 48 h of culture. Thus, substance P may play a role in potentiating matrix deposition in vivo when combined with TGFβ, although this potentiation may be dependent on the concentration of each factor. Treatments targeting substance P may be a viable strategy for treating fibrosis where both substance P and TGFβ play roles.

## 1. Introduction

Substance P is a member of the tachykinin family, a group of peptides known for their ability to rapidly stimulate contraction of intestinal muscle. Tachykinin peptides (eledosin, physalaemin, phyllomedusin, amphibian skin peptides, substance P) and related polypeptides are also characterized by a prompt stimulant action on extravascular smooth muscle and hypotensive action in the vasculature, in contrast to slow-acting kinins (bradykinins) [1]. Substance P is an 11-amino acid peptide that is translated from any one of four alternative mRNA transcripts (α, β, γ, δ) that can be transcribed from the TAC1 gene [2]. It is highly expressed in both the central and peripheral nervous system. Substance P is associated with pain and nociceptor signaling in sensory nerves. There are three tachykinin receptors encoding seven membrane-spanning G-protein-coupled receptors: NK-1R, NK-2R, and NK-3R. Substance P has the highest affinity for NK-1R through which it mediates many of its effects [3].

There are a multitude of observations detailing the role of substance P in fibrotic diseases in different tissues, including heart, lung, kidney, liver, and others. For example, after spinal cord surgery, substance P contributes to epidural fibrosis [4]. In the colon, substance P may play a role in chronic inflammation, evidenced in vivo through the neurokinin-1 receptor (NK-1R)-dependent activation of fibroblasts in accumulating at sites of inflammation, fibrosis, and inflammation, and in vitro by its ability to induce an increased expression of intracellular collagen and increased migration of a human-colon fibroblast cell line [5]. On the other hand, in some instances, substance P can play an anti-fibrotic role [6]. For example, in a primate model of type 2 diabetes where cardiac fibrosis is associated with heart failure, treatment with exogenous substance P was beneficial [7]. These contradictions support the need to examine the specific role of substance P in different tissue types and disorders.

Fibrosis is a major factor in musculoskeletal disorders. We and others have demonstrated the occurrence of fibrosis in experimental models of musculoskeletal overuse injury [8,9]. Using a unique rat model of upper extremity repetitive strain injury, we have reported that the long-term performance of a highly repetitive forelimb task results in increased immunoreactivity of both substance P and NK-1R in neurons and axons in spinal cord dorsal horn lamina [10,11], in parallel with increased forepaw sensitivity and cold sensitivity in the rats. Substance P also appears to be an important factor in overuse induced tendinopathy [12,13,14,15]. In vivo, substance P expression is increased in tendons and muscles of animals with repetitive strain injury [14,16,17]. It is also increased in human tendons showing signs of tendinosis, a tendon fibrotic pathologic condition [18]. Also, substance P injected into Achilles tendon paratenon increases inflammation in the paratenon and the number of blood vessels in the tendon proper in a rabbit tendinopathy model [15]. In addition, the in vitro treatment of tenocytes with substance P enhances the production of fibrotic proteins, including collagen I and III [19,20]. In tenocytes, substance P induced collagen type I production, after 2 days of culture, but was blocked by a transforming growth factor beta (TGFβ) receptor inhibitor, suggesting that substance P was acting through the TGFβ pathway. Owing to its increased expression after tendon rupture and subsequent healing [21] and in contrast to its pathological role in tendinosis, substance P has been proposed as a mediator of healing and recovery after tendon rupture. Subsequently, the injection of substance P into the site of tendon rupture has been shown to be beneficial during healing [22,23].

Since substance P and NK-1R are elevated in models of overuse and in human overuse pathologies, the substance P pathway has been proposed as a therapeutic target for musculoskeletal disorders [24]. In addition to its role in nociception, substance P may also play a role in the development of fibrosis in musculotendinous tissues. We have shown an increase in substance P in tendon [16], muscle, and skin [17] associated with fibrosis in our model of forelimb overuse injury. Using this model of repetitive injury, we examined whether treatment with a specific human NK-1R antagonist (NK-1RA), L732,138, would prevent the development of fibrosis in musculotendinous tissues [25]. Rats performing a 3-week, high-force, high-repetition, lever-pulling task were untreated or received treatment in task weeks 2 and 3 with the NK-1RA. The treatment reduced HRHF-induced thickening in involved epitendons and production of profibrotic factors including TGFβ1 and collagen I in muscles. The NK-1RA also reduced task-induced increases in collagen deposition in the dermis of involved forepaws.

In skin, a role for substance P in chronic inflammation that leads to fibrosis has been proposed. There is in vitro evidence that human dermal fibroblasts express NK-1R [26]. In response to substance P exposure, these cells express mRNA from the TAC1 gene and substance P protein in response to substance P exposure [27]. In vitro evidence also demonstrates that substance P can mediate some biological effects on fibroblasts derived from skin. Parenti A. et al. (1996) showed that NK-1R stimulation increases the migratory behavior of human dermal fibroblasts [28], while NK-2R and the NK-3R agonists had no effect. Substance P may also induce a TGFβR-dependent proliferation of human dermal fibroblasts [29]. This is pertinent because TGFβ is a key central mediator of fibrosis [30] in vivo and is involved in many fibrotic diseases and disease models.

Since NK-1R activation is implicated in fibrosis in several tissues, including skin, we sought to investigate the hypothesis that substance P would increase fibrotic tendencies in rat dermal fibroblasts. We investigated its ability to affect the proliferation and differentiation of rat primary dermal fibroblasts, as well as their secretion and deposition of collagen secretion and deposition in short- and long-term culture experiments. In addition, in light of prior in vivo and in vitro experiments demonstrating that some fibrotic effects of substance P may be induced indirectly through the TGFβ signaling pathway [19], we also compared its effects with those of TGFβ alone, or in combination with TGFβ, in both short- and long-term in vitro cultures of rat dermal fibroblasts.

## 2. Results

### 2.1. Proliferation

Proliferation was measured by assessing the number of BrdU-positive nuclei present in cell cultures from 48-well plates using flow cytometry (Figure 1A). To detect BrdU, DNA must be denatured to facilitate access of the anti-BrdU antibody to the incorporated BrdU. We used an acidic treatment to accomplish this. We did not observe significant fluorescence and thus the incorporation into newly synthesized DNA in cells not treated with acid.

When cells were stimulated in the presence of 10% FCS, there was substantial cell proliferation and DNA synthesis, even in untreated control cells (Figure 1B,C). There were decreased total numbers of cells (assessed by counting DAPI-stained nuclei) in cultures after 48 h of TGFβ treatment (5 ng/mL), with or without cotreatment with substance P (100 nM) (Figure 1B), as well as lower percentages of BrdU-positive cells (Figure 1A,C). (we use ng/mL instead of nanomolars in reference to TGFβ concentration for reasons of comparison with our previous publications and many other publications available in the literature; with TGFβ having a molecular weight of 25 kilo Daltons, 5 ng/mL is equal to 200 pM of TGFβ). In contrast, cells treated with only substance P had similar numbers of cells and a similar BrdU incorporation as untreated control cells (Figure 1B,C).

We next examined the long-term effect of TGFβ (5 ng/mL) and substance P (100 nM) treatments on cell numbers in the RDF cultures (Figure 2). We stained nuclei in day 5 cultures with DAPI and day 12 cultures with Hoechst 33342 and counted the fluorescent images using ImageJ (version 2.0.0-rc-69/1), as described in the methods. Representative day 12 images are shown in duplicate in Figure 2A. Data from days 5 and 12 give a similar pattern of results (Figure 2B and Figure 2C, respectively). At each time point, there was a significant decrease in the numbers of cells in cultures treated with either TGFβ alone, or TGFβ and substance P combined, compared to untreated control cells and the substance P only treatment. Substance P did not significantly change the cell numbers, compared to untreated control cells, although at day 12, there was a small yet insignificant increase in cell numbers (*p* = 0.07; Figure 2B,C).

A three-way ANOVA indicated there were significant variations, with the TGFβ treatment giving 76.9% of the total variation, while the substance P treatment provided less than 1% of the total variation. The independent cell culture factor provided 4.5% of the total variation (Table 1), while the combination of substance P and TGFβ provided a substantial variation (2.5–5.3% of total).

### 2.2. Soluble Collagen I in Conditioned Medium

Since the amount of collagen I secreted into the medium of any well may be dependent on the number of cells in the well, especially since the latter differed with treatments, we normalized the soluble/secreted amounts of collagen I (detected using ELISA) by dividing the collagen type I concentration in a plate well by the representative number of cells counted in that well (Figure 3). For the 0–5 day time interval (Figure 3A), there was no statistical difference in the amount of secreted collagen I per cell in the media of any treated cell group compared to untreated cells. That said, there was less secreted collagen I per cell in the medium of substance P-treated cells (100 nM) compared to the medium of cells treated with TGFβ and substance P combined (5 ng/mL and 100 nM, respectively). At the later time interval of 9–12 days (Figure 3B), there was less secreted collagen I per cell in the media of all treated groups compared to the untreated cells.

### 2.3. Matrix Collagen Deposition

While we expected the amount of collagen deposition to increase with TGFβ treatment [32,33,34], the effects of substance P alone or in combination with TGFβ on cultured dermal fibroblasts on collagen deposition was unknown. Thus, we next examined the effect of these treatments on matrix collagen deposition (representative images are shown in Appendix A). We stained dermal fibroblast cultures in the wells of 48-well plates with the collagen-specific picrosirius red stain that specifically stains collagen [35] and used Image J software [31] to analyze the brightfield images of the cultures. We measured the total absorption in four different rat dermal fibroblast cell cultures at the different time points, as described in the Section 4. Again, we normalized the data to numbers of cells per well since the treatments affected cell numbers.

Absorbance data quantifying the amount of collagen deposited in the cultures at day 5 and day 12 normalized to cell numbers is presented in Figure 4. There was no significant difference between the substance P (100 nM) only treated cells and the untreated cells. However, at both time points, there was increased matrix collagen deposition in 5 ng/mL TGFβ-treated cells compared to untreated cells (Figure 4A,B). Interestingly, at both time points, substance P enhanced TGFβ induction of matrix collagen deposition compared to TGFβ treatment alone (Figure 4A,B).

### 2.4. Concentration-Dependent Effects of TGFβ and Substance P on Fibroblast Cell Numbers

Since this suggests that substance P could have a potentiating effect when combined with TGFβ at the concentrations used to this point in our experiments (100 nM and 5 ng/mL, respectively), we further investigated the relationship between these two proteins. We performed these next experiments in 96-well plates to facilitate efficacy and employed an automated system to image each well.

We first performed a wide-ranging titration of both substance P and TGFβ individually in cultures grown and treated across 6 days. We tested TGFβ concentrations from 2 ng/mL (2 × 10^−9^ g/mL) to 0.000000002 ng/mL (2 × 10^−18^ g/mL) and substance P concentrations from 200 nM (2 × 10^−7^ M) to 0.0000002 nM (2 × 10^−16^ M) (Figure 5 and Figure 6, respectively). We found that TGFβ treatment between concentrations 2 × 10^−17^ and 2 × 10^−12^ g/mL increased fibroblast cell numbers after 6 days, significantly for concentrations 2 × 10^−12^, 2 × 10^−14^ and 2 × 10^−16^ g/mL of TGFβ, compared to untreated control cells for one culture tested (Figure 5D). This increase did not reach statistical significance for the other RDF culture tested, although the same trend was observed (Figure 5A). Yet, higher concentrations of 2 × 10^−10^–2 × 10^−9^ g/mL (i.e., between 0.2 ng/mL and 2 ng/mL) of TGFβ had an inhibitory effect on cell proliferation. This inhibitory effect on cell numbers reached statistical significance at the highest concentration tested, 2 × 10^−9^ g/mL (2 ng/mL), for both rat dermal fibroblast cell cultures tested (Figure 5A,D).

The total amounts of matrix collagen deposited per wells did not vary compared to the untreated controls (Figure 5B,E). In contrast, when the data were normalized to the number of cells per well, there was a dose-dependent increase in the amount of collagen deposited per cell at the highest concentrations of TGFβ tested, although this did not reach statistical significance (Figure 5C,F). This trend toward an increase with higher concentrations of TGFβ was very consistent in the two experiments shown (Figure 5C,F).

We also tested the effect of substance P treatment at different concentrations. Substance P treatment for 6 days gave statistically significant higher numbers of cells at concentrations of 2 × 10^−10^ M and 2 × 10^−9^ M (0.2–2.0 nM) for one cell culture (Figure 6A) but not for the second cell culture tested, although there was a similar trend (Figure 6D). There were no differences in the amount of matrix collagen per well (Figure 6B,E) or the amount of matrix collagen per cell in untreated cells versus any concentration of substance P-treated cells (Figure 6C,F).

### 2.5. Substance P Enhances TGFβ-Induced Matrix Collagen Deposition in a Concentration-Dependent Manner

Since Figure 4’s results suggested a potentiating effect of substance P when combined with TGFβ for the induction of collagen deposition, we next tested the effect of four different concentrations of substance P when combined with a high concentration of TGFβ. In one series of experiments, we used 5 ng/mL of TGFβ that we initially discovered to have an augmentation effect with substance P on matrix collagen deposition (from Figure 4). We conducted two experiments using two RDF cultures from passage 2 cells, combined the data, normalized the results to the mean of the untreated cells, and evaluated the results regarding the ability of substance P to increase cell proliferation or collagen production on its own versus in combination with TGFβ at 5 ng/mL and four concentrations of substance P from 2 × 10^−7^ M to 2 × 10^−16^ M were used as shown in Figure 7.

As expected from Figure 2’s and Figure 6’s results, the TGFβ-only treatment reduced the numbers of cells per well, compared to untreated results (dotted versus dashed lines in Figure 7A, *p* < 0.0001). In contrast, the substance P-only treatment significantly increased the numbers of cells per well at the concentrations of 2 × 10^−10^ M and 2 × 10^−13^ M compared to untreated cells (*p* < 0.01 each; asterisks in Figure 7A). The substance P-only treatment also significantly increased the numbers of cells per well at all concentrations compared to the TGFβ-only treatment (*p* < 0.01 for each substance P concentration; not depicted for the simplification of Figure 7A). Importantly, there were no differences in cell numbers per well with any concentration of substance P when combined with TGFβ at 5 ng/mL compared to the TGFβ-only treatment (*p* < 0.01 for each; not depicted for the simplification of Figure 7A). Thus, the TGFβ treatment reduced the numbers of cells per well, with or without substance P cotreatment.

Regarding matrix collagen deposition per cell, as expected from Figure 4, the TGFβ-only treatment increased the amounts of deposited matrix collagen per cell compared to untreated results (dotted versus dashed lines in Figure 7B, *p* < 0.0001). In contrast, the substance P-only treatment at any concentration tested had no effect on the amounts of matrix collagen deposited per cell compared to untreated results (*p* < 0.01 for each substance P concentration; not depicted for the simplification of Figure 7B). Yet, the amounts of matrix collagen deposited per cell were higher with any concentration of substance P when combined with TGFβ at 5 ng/mL compared to untreated results alone (*p* < 0.01 for each substance P concentration; not depicted for the simplification of Figure 7B). Lastly, and importantly for depicting potentiation, there was also a clear increase in the amounts of matrix collagen deposited per cell when TGFβ was combined either with the 2 × 10^−13^ M or 2 × 10^−10^ M concentration of substance P compared with the TGFβ-only treatment (number symbols in Figure 7B). Thus, TGFβ when combined with substance P at two concentrations induced more collagen matrix deposition than TGFβ only.

We next used a lower concentration of TGFβ to stimulate RDFs to determine if this potentiation was concentration-dependent. Based on the titration results for TGFβ shown in Figure 5, we chose a concentration of 20 pg/mL (indicated in Figure 5 at 2 × 10^−11^ g/mL) since it was a concentration that gave results that were not significantly different from untreated cells, was between the concentrations that gave enhancement and inhibition effects on proliferation, and was a lower concentration than those that enhanced collagen deposition per cell. We tested this 2 × 10^−11^ g/mL concentration of TGFβ (20 pg/mL), with and without different concentrations of substance P. The addition of substance P in combination with TGFβ 2 × 10^−11^ g/mL (20 pg/mL) increased the proliferation of two RDF cultures statistically in one experiment at all four concentrations of substance P tested (Figure 8A) compared to TGFβ-only treated cells but only at two concentrations in a second experiment (Figure 8C). Substance P in combination with TGFβ decreased the amount of matrix collagen deposited per cell in one experiment with substance P concentrations of 2 × 10^−16^ M and 2 × 10^−10^ M and in the second experiment with substance P concentrations of 2 × 10^−13^ M and 2 × 10^−10^ M, 2 × 10^−7^ M compared to cells treated only with TGFβ (Figure 8B). Substance P on its own did not significantly increase proliferation compared to untreated cells in either of the two experiments, but it decreased matrix collagen per cell compared to untreated cells at a concentration of 2 × 10^−7^ M in the first experiment and at all four concentrations tested in the second experiment.

### 2.6. α-Smooth Muscle Actin Expression

Increased αSMA is characteristic for the differentiation of proto-myofibroblasts into fully differentiated myofibroblasts [36]. We observed increased αSMA expression in 48 h and 72 h cultures of RDF cells treated with TGFβ, or TGFβ combined with substance P, compared to untreated control cells (Figure 9A,B). Substance P had no effect on αSMA expression at 48 h of culture compared to untreated control cells yet showed an increase at 72 h compared to untreated control cells (Figure 8A versus Figure 8B). TGFβ treatment increased αSMA expression at both time points compared to untreated control cells, whether the treatment was alone or combined with substance P. Therefore, substance P can increase αSMA expression, although after more time in culture and not as strongly as TGFβ. In addition, at the 72 h time point, substance P when combined with TGFβ reduced the expression of αSMA compared to TGFβ alone.

### 2.7. Expression of Neurokinin-1R in Dermal Fibroblasts

Most of the effects of substance P are mediated through its interaction with the neurokinin-1 receptor, although substance P can mediate some effects through neurokinin-2 and neurokinin-3 receptors. We wanted to assess the expression of the neurokinin-1 receptor in rat dermal fibroblasts. Using a polyclonal antibody that gave the expected positive staining in the spinal cord of rats, we could detect NK-1R-specific fluorescence in fibroblasts. Flow cytometry was used to more easily detect small differences in fluorescence. Our experiments showed that the expression of the NK-1 receptor protein was low and somewhat variable in three different cell cultures of RDFs. (Figure 10A–D). When we combined the data, the average effect of TGFβ appeared to suppress the expression of the NK-1R (Figure 10D).

## 3. Discussion

We investigated the hypothesis that substance P would increase fibrotic tendencies in rat dermal fibroblasts. Since substance P is believed to play a role in fibrosis and at least some of its effects in this arena are thought to be mediated through its ability to activate TGFβ, we sought to examine its independent versus interactive effects on rat dermal fibroblasts in vitro. We compared the effects of substance P alone versus TGFβ alone or in combination with TGFβ. We investigated these effects in short- and long-term cultures of primary rat fibroblasts derived from the skin of adult rats.

In short-term cultures of 2 days, we measured the incorporation of BrdU into newly synthesized DNA to assay proliferation (Figure 1). We found that RDF cells treated with 5 ng/mL TGFβ, with or without the substance P (100 nM) cotreatment, showed less DNA replication and lower cell numbers than the untreated control cells. To measure proliferation in longer term experiments of 5 and 12 days, we counted nuclei to estimate the number of cells in the cultures. We found in these rat dermal fibroblast cultures that 5 ng/mL of TGFβ, either alone or in combination with substance P at 100 nM (1 × 10^−7^ M), consistently decreased the number of cells in the cultures, compared to untreated control cells. However, these results do not rule out the possibility that substance P at lower concentrations could induce low amounts of TGFβ in the fg/mL range that may also induce proliferation, which would agree with previously published data [37].

Therefore, in 6-day cultures, we examined the effects of titrating the concentrations of substance P and TGFβ alone. We found the proliferative effect of substance P was more pronounced at lower concentrations (2 × 10^−10^ M and 2 × 10^−13^ M (i.e., 0.2 nM and 0.0002 nM), although the effect was variable (Figure 6). In contrast, our results in both the short-term with 5 ng/mL TGFβ (Figure 1) and long-term cultures with both 5 ng/mL and 2 ng/mL (Figure 2 and Figure 5, respectively) show cell number inhibition, while concentrations of TGFβ less than 0.0002 ng/mL induced proliferation instead (Figure 5D). The regulation of cell proliferation by TGFβ is highly context specific. TGFβ was first found in murine sarcoma transformed cancerous cells and chemically transformed cells from the bladder and trachea [38,39] and later in normal tissues [40] and was shown to induce proliferation of normal primary fibroblasts cells in soft agar [40]. However, later reports showed that TGFβ could inhibit the proliferation of other types of both normal and cancer cells [41]. In the case of fibroblasts, TGFβ inhibits the proliferation of fibroblasts cultured in 10% FCS when at a concentration of 100 ng/mL [41,42,43], consistent with high concentrations of TGFβ inhibiting proliferation as shown here.

TGFβ is also able to stimulate the production of collagen and its incorporation into extracellular matrix [44]. Since we were also interested in examining the effect of substance P on collagen production and deposition, we first examined the effect of substance P and TGFβ on the amounts of secreted collagen I in the media of cultured fibroblasts. For this, we normalized the data to cell numbers per well since although the initial cell density was the same for all treatments, the treatments had a profound effect on cell numbers present in the cultures across time. In general, the amounts of collagen I secreted into the media increased over time after normalization to the number of cells per well of the culture plates, even for untreated cell cultures (Figure 3A versus Figure 3B). At the 0–5-day time interval, the greatest amount of collagen secretion was by TGFβ (5 ng/mL) and substance P (100 nM) combination-treated cells, compared to TGFβ alone (Figure 3A). This difference was lost by the 9–12 day interval in which all treatments induced less collagen secretion than that by untreated cells (Figure 3B). These data suggest that the combined treatment of substance P with TGFβ enhances collagen secretion more than either treatment alone in rat dermal fibroblast cultures in a time-dependent manner (shorter-term cultures but not longer-term). Our results differ from previous results showing that tenocytes secreted more collagen I after substance P treatment (with or without TGFβ cotreatment) compared to TGFβ alone at 24 h of culture, yet, similar amounts of collagen secretion after treatment with substance P, TGFβ, or substance P and TGFβ cotreatment by 48 h of culture [19]. The discrepancy could be due to differences in total culture times (9–12 days in this current study versus 48 h in the tenocyte study) or different culture conditions as the tenocyte experiments used media with low 1–2% serum compared to the 10% serum we used for these dermal fibroblast experiments. The discrepancy might also reflect fundamental differences between dermal fibroblasts and tenocytes in their ability to respond to substance P. Interestingly, the tenocyte culture study [19] showed that substance P potentiated TGFβ production, particularly at 48 h of culture, and that substance P production of collagen at 48 h of culture was via the TGFβ signaling pathway (it was blocked with a TGFβ receptor inhibitor).

Also, however, the amount of soluble collagen I in the medium may not reflect the amount of collagen I secreted by the cells, since most collagen I is incorporated into the extracellular matrix and therefore is not soluble. Therefore, we examined the effects of substance P and TGFβ on the deposition of collagen into the extracellular matrix by the cultured rat dermal fibroblasts. Using microscopy, we measured the light absorption of dermal fibroblasts stained with a collagen specific dye, picrosirius red combined with picric acid (Figure 4 and Appendix A), as an estimate of total collagen deposited in the individual cultures. Then, collagen deposition was evaluated as a per cell value, by normalization to cell number as described for soluble collagen I. On both days 5 and 12, there was no change in collagen deposition in response to substance P treatment compared to untreated control cells. Yet, TGFβ treatment increased collagen deposition compared to the untreated control cells. Furthermore, the combination of TGFβ and substance P resulted in a significant increase in collagen deposition compared to TGFβ alone at both time points, a result that has not been previously reported to our knowledge. The calculation of this potentiating effect at day 5 and day 12 for the individual cultures is shown (Appendix A).

The results showing that TGFβ treatment at 5 ng/mL in isolation stimulates collagen deposition per cell is in agreement with the direct effect of TGFβ on matrix collagen deposition that has been previously described [30,44] and the multitude of reports showing that TGFβ is strongly associated with pathogenic fibrosis [45,46,47,48]. Our results extend the literature to now show that substance P also plays a role in enhancing the deposition of collagen and extracellular matrix in situations where there is ongoing TGFβ-mediated stimulation of ECM production.

We then tested different concentrations of substance P in combination with the high concentration of TGFβ that in our initial experiments had shown a potentiating effect for collagen deposition (Figure 7 versus Figure 4). We were able to demonstrate the potentiation of matrix collagen deposition per cell in our experiments with lower concentrations of substance P (Figure 7B). We observed an increase in matrix collagen deposited per cell when 5 ng/mL TGFβ was combined with either the 2 × 10^−13^ M or 2 × 10^−10^ M concentration of substance P (0.0002 and 0.2 nM) compared with 5 ng/mL of the TGFβ-only treatment. Thus, TGFβ when combined with substance P at fairly low concentrations induces more collagen matrix deposition than TGFβ only. We also tested the effects of a lower concentration of TGFβ (20 pg/mL), with and without substance P. In these experiments substance P on its own did not significantly increase proliferation but it did decrease slightly matrix collagen deposition per cell. In combination with the lower concentrations of TGFβ, it did modulate the slight inhibitory effect of TGFβ on proliferation and the slight positive effect of TGFβ on matrix collagen deposition. Thus, the lower concentration of TGFβ combined with substance P did not have a potentiating effect with regard to collagen matrix deposition; therefore, this augmentation of matrix collagen production when combining TGFβ and substance P is dependent on the concentration of each factor.

We tested two different concentrations of TGFβ with several concentrations of substance P to demonstrate the positive potentiation of collagen deposition. Considering the proliferative effect of lower concentrations of TGFβ, other concentrations of TGFβ, especially lower concentrations, could be tested to further investigate the interaction of the two proteins.

In many models of fibrosis and in fibrotic tissues, the differentiation of fibroblasts into myofibroblasts has an important role in the pathogenic process of fibrosis [49]. αSMA is a well-accepted marker of smooth-muscle cells and is used as a marker to identify fibroblasts that have differentiated into myofibroblasts [50]. Myofibroblasts contribute to the contraction of healing tissue [43] and the resolution of wound healing. TGFβ, as well as inflammatory cytokines, induce the accumulation of αSMA-positive fibroblasts [51]. Our results suggest that while substance P alone may have some effects on fibroblasts including increasing SMA expression after 3 days of culture (Figure 8) and increasing their proliferation, substance P does not increase collagen secretion or deposition in the absence of TGFβ (Figure 4, Figure 6 and Figure 7).

We also examined the expression of NK-1R, the high affinity receptor for substance P, in the rat dermal fibroblasts. Our flow cytometry data demonstrated low expression levels of NK-1R on the surface of the rat dermal fibroblasts, while stimulation with TGFβ tended to reduce the expression of NK-1R. These data suggest that the effects of substance P on rat dermal fibroblasts can be mediated through ligation of the NK-1R expressed on the cells. However, substance P effects can also be mediated through the other neurokinin receptors, NK-2R and NK-3R. Future experiments to address these possibilities are planned.

The system we used was one of long-term primary cell cultures whose biological properties change over time. The presence of 10% fetal bovine serum in the culture media introduced a complex mixture of growth factors that can affect individual cell cultures differently. Also, the long-term nature of most of the experiments led to a substantial variation in our results. We were nevertheless able to uncover some important information regarding the effects of substance P on rat dermal fibroblasts. Our results indicate that substance P on its own can positively affect the proliferation of rat dermal fibroblasts, but only in the presence of TGFβ does it have a positive stimulatory effect on the deposition of matrix collagen. We found that the quantity of collagen in the conditioned medium of fibroblasts is not a good measure for evaluating the fibrotic tendency of fibroblasts in long-term experiments in which collagen is deposited into the extracellular matrix. In vitro collagen deposition may be a better parameter to predict the potential for fibrosis in vivo. Importantly, we show for the first time that increases in collagen deposition in cultures exposed to both substance P and TGFβ for 5 to 12 days, compared to either treatment alone, substance P enhances the ability of TGFβ to induce matrix collagen deposition, suggesting that substance P may play a supplementary role in pathological fibrosis but only in the presence of high concentrations of TGFβ. Therefore, targeting the substance P pathway may be a valid therapy to inhibit substance P enhancement of TGFβ-mediated fibrosis in conditions where both factors are playing profibrotic roles.

## 4. Materials and Methods

### 4.1. Animals

All experiments were approved by the Temple University Institutional Animal Care and Use Committee (Temple University IACUC-approved protocols 5054 and 4787) in compliance with NIH guidelines for the humane care and use of laboratory animals. Female Sprague–Dawley young adult rats were used. These rats were housed in the ULAR animal facility and maintained in accordance with Temple University Laboratory Animal resource guidelines (including being housed in a central animal facility in a 12 h light: 12 h dark cycle with free access to water). All rats were handled at least twice per week and provided cage enrichment toys that included chew bones, tunnels, and paper twists (Diamond Twists, Envigo, South Easton, MA, USA, Teklad 7979C.CS) until tissue collection.

### 4.2. Primary Fibroblast Cultures

Adult rat dermal fibroblasts (RDFs) were prepared from underarm skin from four 4–8-month-old adult rats that were maintained as different independent primary cell cultures. For this, the underarm area was shaved and sterilized with 70% ethanol. An area measuring approximately 1 cm^2^ was dissected. Skin RDFs were prepared from skin using a modification of a published protocol [52]. Dissected skin tissue was minced with two scalpels and digested with Liberase (Roche, obtained from Sigma-Aldrich, St Louis, MO, USA, Catalog # 05401119001) in 5 mL of DMEM/F12 medium at a concentration of 0.14 Wunsch units/mL in a sterile 50 mL Erlenmeyer flask for 30 min at 37 °C with orbital shaking. The solution with skin tissue was pipetted up and down to break up the tissue and then transferred into a 15 mL conical tube. The flask was rinsed with medium containing 15% fetal calf serum (FCS; BioWest, Bradenton, FL, USA, Catalog # S152I) and penicillin and streptomycin (Penn/Strep; Mediatech, Manassas, VA, USA, Catalog # 30-002-CI), and the rinse was added to the conical tube. The digested tissue was centrifuged at 400 g in a swinging bucket, and the tissue pellet was washed again with the medium containing 15% FCS. The tissue was resuspended in the medium and divided into two 10 cm tissue culture dishes with 10 mL of medium in each dish and cultured for 3–5 days until adherent cells were seen proliferating in the dish. The medium was removed carefully without disturbing the tissue and replaced with fresh medium containing 15% FCS. Cells were cultured to confluence and were trypsinized and subcultured and the remaining cells were frozen down in freezing medium for storage in liquid nitrogen. The subcultured RDFs were maintained in medium containing 15% FCS before being prepared for use in experiments. At that point, RDF cells were cultured in medium containing 10% FCS, with or without 100 nM of substance P (Tocris Bioscience, Bristol, UK, Catalog # 1156) and/or 5 ng/mL of human TGFβ1 (Peprotech, Rocky Hill, NK, USA, Catalog # 100-39) (we used human TGFβ because unlike the rat protein it is readily available, has almost 100% identity with rat TGFβ, and has similar effects on rat cells as it does on human cells.) Medium was exchanged 3 times/week until the end of the culture period. We also tested titrated concentrations of TGFβ and substance P. For these experiments, we started with concentrations of 2 ng/mL for TGFβ and 200 nM for substance P and used a vol:vol, 1 protein:9 (medium) titration series to construct a concentration series for testing.

### 4.3. Fibroblast Proliferation

Proliferation was quantified either by measuring bromodeoxyuridine (BrdU) incorporation into newly synthesized DNA using flow cytometry for experiments from 1–3 days of culture or by counting cell numbers employing DAPI- or Hoechst 33342-stained nuclear counts. Fibroblasts were plated to give a cell concentration of 8000 per cm^2^ in 48-well plates in MEM medium (GE Healthcare Services, Logan, UT, USA, Catalog # SH30024.01) containing penicillin/streptomycin (Penn/Strep), L-glutamine (MP Biomedicals, Solon, OH, USA, Catalog # 1680149), nonessential amino acids (GE Healthcare Services, South Logan, UT, USA, Catalog # SH30238.01), and sodium pyruvate (GE Healthcare Services, South Logan, UT, USA, Catalog #SH30239.01) with added 1% FCS. In later experiments, to facilitate shorter culture periods, cells were plated at 16,000/cm^2^ in 96-well plates. Cells were cultured for 1–3 days in this medium to arrest growth in the G0/G1 phase of mitosis. The medium for experiments was MEM containing 10% FCS and with additives Penn/Strep, L-glutamine, nonessential amino acids, and sodium pyruvate, as described above, as well as 50 µg/mL of ascorbic acid (Sigma, St. Louis, MO, USA, Catalog # A0278). Initial experimental treatments were in the presence or absence of 100 nM of substance P (Tocris, Bristol, UK, Catalog # 1056) and/or 5 ng/mL of human TGFβ1 (Peprotech, East Windsor, NJ, USA, Catalog # 100-21-C). Proliferation was assessed by BrdU incorporation in fibroblast cultures, using a previously described method [53]. Briefly, 50 mm BrdU was added to cell cultures 4 h before they were trypsinized and harvested for flow cytometry analysis. Cells were fixed overnight or longer in 70% ethanol, washed sequentially in phosphate buffered saline (PBS), and then in 1 N HCl before being treated with 2 N hydrochloric acid containing 0.1% Triton-X-100 for 20 min at room temperature. The 2 N acid was removed by centrifuging, removing the supernatant, and washing several times in 0.2 M Tris pH 7.5 until the supernatant pH reached neutrality. Cells were then stained in PBS containing 0.5% bovine serum albumin with anti-BrdU antibody conjugated to Alexa 488 (clone 3D4, Biolegend, San Diego, CA, USA, Catalog # 364106) or allophyocyanin (APC; BD Pharmingen, San Diego, CA, USA, 51-23618L). Cells were washed in PBS and resuspended in 400 µL of PBS containing 1 µg/mL 4′, 6 Diamidino-2-phenylindole dihydrochloride (DAPI; Sigma, St Louis, MO, USA, Catalog # D9542). Data were collected using an LSRII flow cytometer (BD Biosciences, Franklin Lakes, NJ, USA) at a maximum flow rate for 2 min and then analyzed using Flowjo version 10.10.0, Ashland, OR, USA). The total single cell count was used for the relative cell number for each sample.

#### 4.3.1. Hoechst Live Cell Staining

Hoechst 33342 (Molecular Probes, Eugene, OR, USA, Catalog # H1319) was added to cells cultured in duplicate wells of 48-well plates at a concentration of 0.5 ng/mL for the final 2 h of the culture period. The 100× images were obtained on an inverted microscope (Nikon TE E300, Nikon, Melville, NY, USA) and a Nikon digital camera and software (NIS-elememts, Version F). Nuclei were counted using Image J version 2.0.0-rc-69/1.51h software to threshold the nuclei by adjustment of the brightness, conversion to binary image, and then automatically counted.

#### 4.3.2. DAPI Fixed Cell Staining

Cells in 48-well plates were stained with picrosirius red, as described below. After staining, cells were imaged for collagen deposition. They were then destained using multiple changes of acidified water, washed twice in PBS, permeabilized in 0.1% triton-X-100 in PBS, and stained with 1.0 μg/mL DAPI (in 0.1% triton-X-100). The 10× images were obtained from the center of each well of the 48-well plates using an inverted microscope and an appropriate cube for imaging DAPI staining nuclei. Nuclei were counted by thresholding fluorescence to isolate the nuclei and render the image to binary and count the identified nuclei using the analyze particle feature in NIHH Image J version 2.0.0-rc-69/1.51h. The number of nuclei per well was estimated as the cell number in the image from the well center, multiplied by 9, since the diameter of the well was 3 times the diameter of the field of view. This number obtained was used to normalize the soluble collagen and deposited collagen data (quantification methods of collagen are described below). Cells cultured in 96-well black-walled plates (Costar, 3603 Corning Inc. Corning, NY, USA) were fixed and stained with DAPI. Nuclei were counted by imaging the entire wells using an EVOS automatic microscope and counting DAPI-stained nuclei using Image J, as described above.

### 4.4. Alpha-Smooth Muscle Actin (αSMA) Flow Cytometry

αSMA was quantified by flow cytometry in cultures, with or without 5 ng/mL of TGFβ, and/or 100 nM of substance P. Cells were trypsinized, fixed in 70% ethanol, and treated with acid, as described for BrdU staining above. 50 µL of anti-αSMA-Alexa 488 (abcam, Burlingame, CA, USA, Catalog # ab202295) at a dilution of 1:500 was added to the cell pellets after neutralization. Data were collected on the LSRII cytometer and analyzed using Flowjo software version 10.10.0.

### 4.5. Collagen I ELISA

Collagen I was quantified in duplicate in cell culture supernatants collected at experimental time points specified in the Results section. Culture medium containing 5 ng/mL of TGFβ and 100 nM pf substance P were used in the experiments. A rat collagen I ELISA kit from LSBio (Seattle, WA, USA, Catalog # LS_F5638) was used according to the manufacturer’s protocol. Collagen amounts in the medium were normalized to cell number by dividing the collagen I concentration as detected by ELISA obtained in each well with the cell number counted in a field of view from that same well.

### 4.6. Picrosirius Red Staining and Quantification

#### 4.6.1. Microscopic Quantification

Picrosirius red staining for collagen assessment was modified from previously described protocols [54,55]. Cells were fixed using 4% buffered paraformaldehyde for 30 min. The picrosirius red solution was prepared by dissolving sirius red (Direct Red 80, Sigma, St Louis, MO, USA, Catalog # 365548) in a saturated solution (1.3%) of picric acid (Sigma, St Louis, MO, USA, Catalog # P6744). Picrosirius red solution was added to the plates to cover the cells (100 μl/well in 48-well plates) and incubated at room temperature for 30 min. The picrosirius red solution was rinsed off with 3 changes of acidified water (0.5% Acetic acid), and the adherent cells were stored in PBS before imaging. Cells were imaged on an inverted microscope in brightfield. The staining was quantified by imaging each well and then measuring total absorbance using Image J software version 2.0.0-rc-69/1.51h [31]. Absorbance was calculated by subtracting the measurements of each well from that of a control well with no staining. Collagen deposition was normalized to cell number by dividing the absorbance value by the cell number.

#### 4.6.2. Solubilization of Sirius Red Plate Reader Quantification

In addition, in the 96-well plate culture experiments, collagen was quantified by measuring solubilized picrosirius red absorbance using a modification of the method described previously [55]. A standard curve of purified collagen I (Gibco A10483-01 obtained from ThermoFisher, Waltham, MA USA) was used as a standard. Appropriate dilutions of the collagen I solution were made so that aliquots of 10 mL each pipetted into the wells of a 96-well plate in duplicate gave collagen amounts of 5, 3, 2, 1, 0.8, 0.4, 0.2, and 0.1 mg/10 mL. Duplicate samples of each solution were warmed at 40 °C until completely dried. The samples were then fixed in 4% paraformaldehyde and stained in the same manner as described above for cells. After picrosirius red staining, as described above, the plates were washed twice in deionized water, 220 mL of 0.1 N NaOH was added to each well, and the plates were held at room temperature for 2 h or at 4 °C overnight. The plate was washed and was shaken for 1 min at 300 rpm and then 200 mL from each well was removed to a new 96-well plate. Absorbance was measured at 540 nm.

### 4.7. Immunofluorescent Detection of Neurokinin-1R in RDF Cultures

#### 4.7.1. Microscopy

Neurokinin-1 receptor was detected using a polyclonal anti-neurokinin 1 antibody. Cells in 96-well plates were fixed with Bouin’s fixative for 6 h at 4 °C. Cells were blocked with glycine blocking buffer (50 mM glycine, 2% goat serum, 0.5% bovine serum albumin, 0.1% triton-x-100, 0.05% Tween 20). The blocking solution was removed and anti-neurokinin-1R rabbit polyclonal antibody (abcam, Burlingame, CA, USA, Catalog # ab5060) 1:500 dilution was detected with anti-rabbit Cy-3 antibody labeled with Cy3 (Jackson Immmunoresearch, West Grove, PA, USA, Catalog number 111-165-144) at 1 mg/mL. Wells not stained with the primary antibody and only with the secondary antibody were used as controls. Cells were imaged using a Revolve digital microscope (Discover Echo Inc., San Diego, CA, USA).

#### 4.7.2. Flow Cytometry

RDFs (0.5 × 106) were seeded into 10 cm dishes for 3 days in 10% FCS-containing medium. Medium was removed and replaced with fresh medium or medium containing 20 pg/mL or 5 ng/mL TGFβ1. After 3 days, cells were trypsinized, washed in PBS, fixed for 2 h in 4% paraformaldehyde, permeabilized in 0.1% triton-x-100 in PBS containing 0.2% bovine serum albumin overnight at 4 °C, and washed twice in PBS. Cells were then blocked in 200 mL of glycine blocking buffer, and 50 mL of cells was then stained with anti-neurokinin-1R rabbit polyclonal antibody (abcam, Burlingame, CA, USA, Catalog # ab5060) at a 1:500 dilution. As controls, cells incubated with normal rabbit IgG 2 mg/mL were used to detect background and nonspecific level-binding of rabbit antibody binding. Cells were incubated overnight at 4 °C and subsequently washed twice in PBS. Cells were then incubated with anti-rabbit Alexa 488 (Jackson Immmunoresearch, West Grove, PA, USA, Catalog number 111-547-003) at 1 mg/mL for 4 h at 4 °C, washed in 1 mL PBS, and resuspended in 400 mL of PBS. Data were analyzed using an LSRII flow cytometer (BD Biosciences, Franklin Lakes, NJ, USA) and analyzed using FlowJo™ oftware for Mac (Version 10.9. Ashland, OR, USA: Becton, Dickinson and Company).

## 5. Statistics

Statistical analysis was performed using Graphpad Prism (GraphPad Prism version 10.10.0). One-way Brown–Forsythe and Welch ANOVA tests followed by Dunnett’s T3 multiple comparison tests or one-way ANOVAs followed by Dunnett’s multiple comparison tests were used for the 96-well plate experiments to compare treated cells with untreated control cells. Two-way or three-way ANOVAs were followed by Tukey’s multiple comparison tests. The Hoechst nuclear-staining relative cell number data were analyzed using a three-way ANOVA. *p* values of <0.05 were considered significant for all comparisons. All data were graphed using Graphpad Prism version 10.10.0.

## Figures and Tables

**Figure 1 ijms-25-01862-f001:**
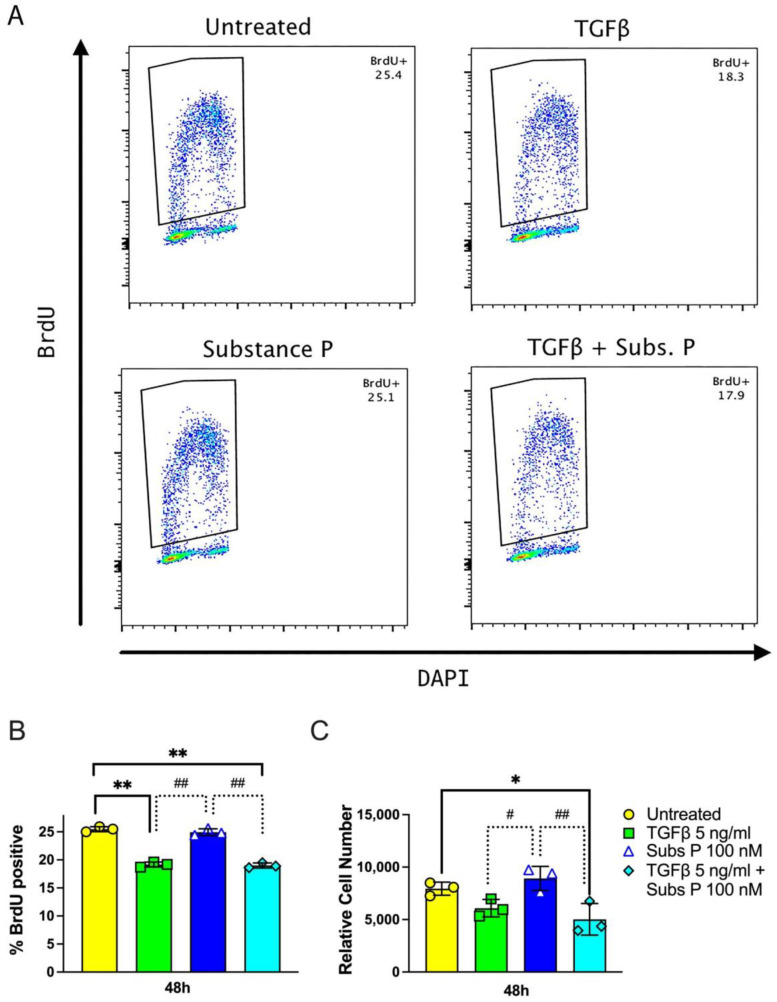
Fibroblast proliferation. Primary dermal fibroblasts cells from an adult rat were cultured with or without 5 ng/mL TGFβ, 100 nM substance P, or with TGFβ (5 ng/mL) and substance P (100 nM) together on 12-well plates for 48 h. BrdU was added in the final 4 h of culture. Cells were harvested, fixed in 70% ethanol, and stained with anti-BrdU and DAPI. (**A**) BrdU-positive cell gates. Single-cell gates were plotted for each condition, and a gate was drawn around BrdU-positive cells. Numbers in the upper right corner of each plot represent the percentage of BrdU-positive cells in the single-cell gates. (**B**) BrdU incorporation. The percentage of cells positive for BrdU incorporation was obtained as described and illustrated in (**A**). (**C**) Relative cell number obtained from flow cytometry data. Data points are presented as mean values with standard deviation error bars. Statistical significance was determined using a two-way ANOVA followed by Tukey’s multiple comparison tests. * *p* < 0.05 and ** *p* < 0.01, respectively, compared to untreated control cells; ^#^ *p* < 0.05 and ^##^ *p* < 0.01, respectively, compared between treated cells as shown.

**Figure 2 ijms-25-01862-f002:**
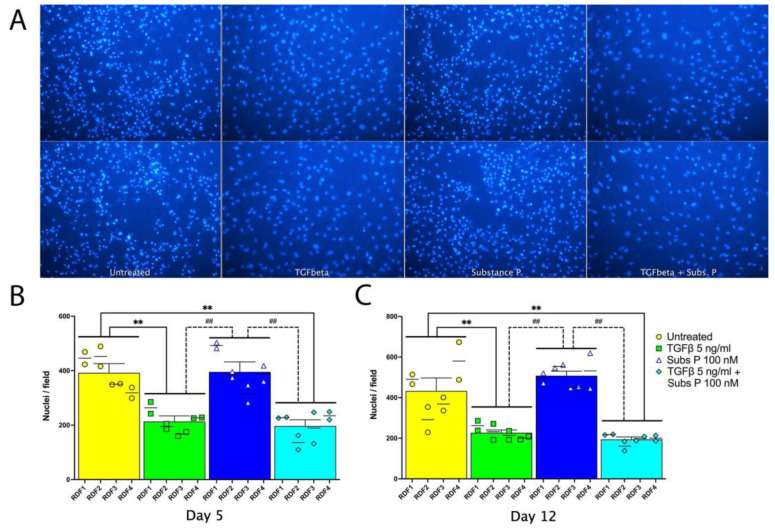
Cell numbers after 5 or 12 days of culture in four independent primary cultures of dermal fibroblasts, obtained from 4 different adult rats. Cells were cultured untreated or were treated with TGFβ 5 ng/mL, substance P 100 nM, or TGFβ (5 ng/mL) and substance P (100 nM) together. There were two replicates per independent culture and each experimental condition. For day 5 cell counts, cells were fixed and stained using DAPI. For day 12 cell counts, live cells were labeled with Hoechst 33342. Cells were imaged using an inverted microscope. Images were converted to binary data by thresholding the DNA stain to identify nuclei, and the nuclei were counted using Image J software version 10.1.0 [31]. (**A**) Hoechst staining of RDF1 cultured for 12 days that were left untreated or treated with TGFβ 5 ng/mL, substance P 100 nM, or both TGFβ (5 ng/mL) and substance P (100 nM). Images were captured using a 20× lens. Upper and lower images are from duplicate wells for each treatment. (**B**) Day 5 cell counts. (**C**) Day 12 cell counts. Data points are presented as mean values with standard error of the mean error bars. Statistical significance was determined using a two-way ANOVA followed by Tukey’s multiple comparison tests. ** *p* < 0.01, compared to untreated cells; ^##^ *p* < 0.01, compared between treated cells as shown.

**Figure 3 ijms-25-01862-f003:**
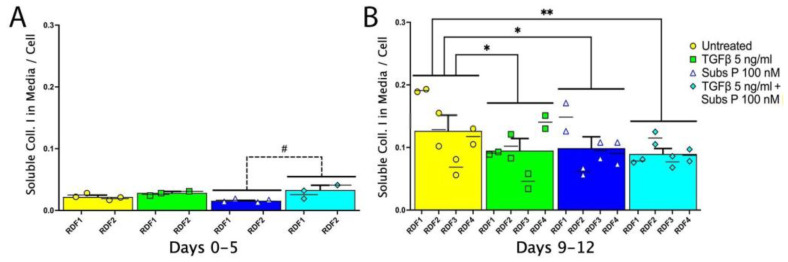
Collagen I secretion measured by ELISA and normalized to cell number. Two independent primary cultures from two different rats were used, in duplicate, for each experimental condition. Cells were cultured untreated or were treated with TGFβ 5 ng/mL, substance P 100 nM, or TGFβ (5 ng/mL) and substance P (100 nM) together before collection of conditioned media at the indicated time points: (**A**) days 0–5 of culture and (**B**) days 9–12 of culture. At each time interval, collagen secretion was normalized to cell number by dividing the collagen I concentration obtained in each well with the cell number counted in a field of view from that same well. Data points are presented as mean values with standard error of the mean error bars. Statistical significance was determined using a two-way ANOVA followed by Tukey’s multiple comparison tests. * *p* < 0.05 and ** *p* < 0.01, compared to untreated cells; ^#^ *p* < 0.05, compared between treated cells as shown.

**Figure 4 ijms-25-01862-f004:**
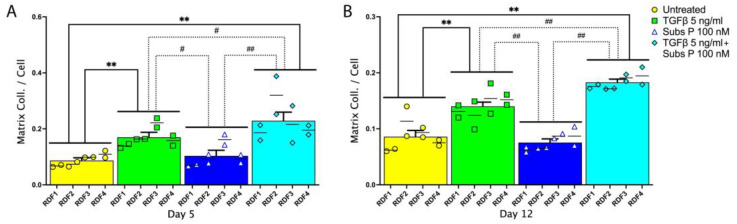
Quantification of collagen deposition in cultures of rat dermal fibroblasts stained with picrosirius red, normalized to cell number. Four independent primary cultures from four different rats were used, in duplicate, for each experimental condition. Cells were cultured, treated as described for Figure 2, fixed in 4% paraformaldehyde, and stained for picrosirius red at the indicated time intervals: (**A**) day 5 and (**B**) day 12 of culture. Images of each well of picrosirius red stained cells were obtained in bright field using an inverted microscope. The picrosirius red stain was quantified as absorbance per well using Image J software version 10.1.0 [31]. We independently collected data from the time points, and these data are dependent on the intensity of the incident light on the cultures, and the absorbance at day 5 gave greater values than that on day 12, but these values are only comparable within each experiment. Collagen deposition was then normalized to cell number by dividing the picrosirius red collagen stain absorption obtained in each well with the cell number counted in a field of view from that same well. Data points are presented as mean values with standard error of the mean error bars. Statistical significance was determined using a two-way ANOVA followed by Tukey’s multiple comparison tests. ** *p* < 0.01 compared to untreated cells; ^#^ *p* < 0.05 and ^##^ *p* < 0.01, compared between treated cells as shown.

**Figure 5 ijms-25-01862-f005:**
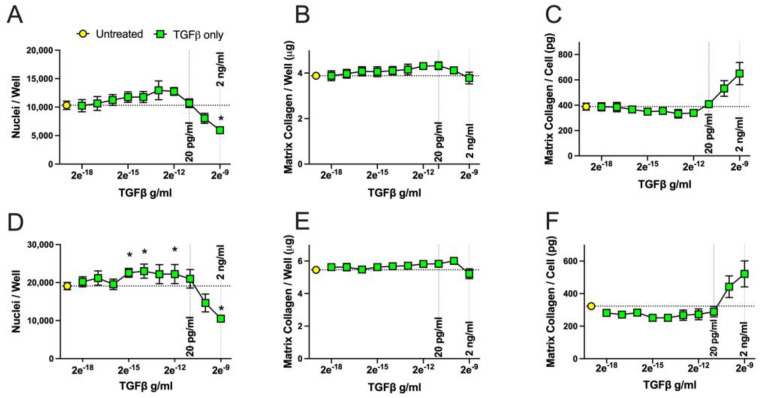
Effect of TGFβ concentration on rat dermal fibroblast proliferation and collagen deposition. Fibroblasts were cultured with an initial density of 16,000 cm^−2^ in 96-well plates. After overnight adherence in 1% FCS-containing medium, cells were incubated for a total of 6 days in 10% FCS-containing medium in the absence or presence of different concentrations of TGFβ, starting at 2 ng/mL (2 × 10^−9^ g/mL) and serially diluted at a ratio of 1:9 to a concentration of 2 × 10^−18^ g/mL). (**A**–**C**) RDF3, (**A**) nuclei/well, (**B**) collagen/well, (**C**) collagen/cell. (**D**–**F**) RDF5, (**D**) nuclei/well, (**E**) collagen/well, (**F**) collagen/cell. Dashed lines on the x axis indicates the concentration of 20 pg/mL that was chosen to investigate the interaction of a lower concentration of TGFβ with substance P and the 2 ng/mL used as the highest concentration of TGFβ in the experiment. The dotted line on the y axis indicates the mean values obtained for the untreated cells. Data points are presented as mean values with standard error of the mean error bars. Statistically significant differences were determined using a Brown Forsyth and Welch ANOVA followed by a Dunnett’s T3 multiple comparison test. * *p* < 0.5, ** *p* < 0.01, compared to the untreated control cells.

**Figure 6 ijms-25-01862-f006:**
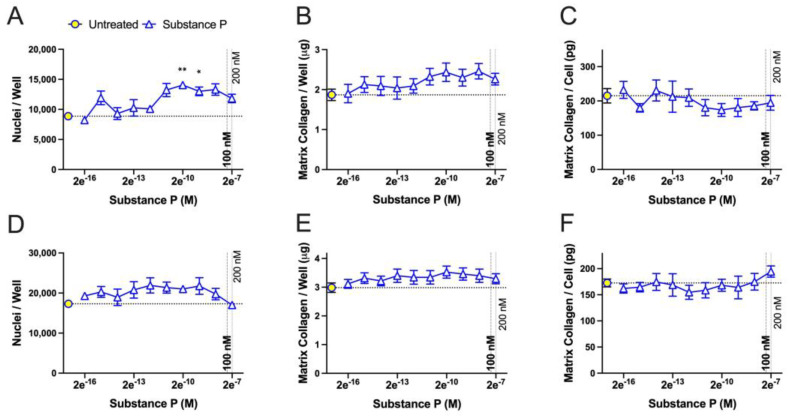
Effect of substance P concentration on rat dermal fibroblast proliferation and collagen deposition after a total of 6 days in culture. Fibroblasts were cultured with an initial density of 16,000 per cm^−2^. After overnight adherence in 1% FCS-containing medium, cells were incubated for a total of 6 days in 10% FCS-containing medium in the absence or presence of different concentrations of substance P starting at 200 μM (2 × 10^−7^ M) and serially diluted at a ratio of 1:9 to a concentration of 2 × 10^−16^ M). (**A**–**C**) RDF3, (**A**) nuclei/well, (**B**) collagen/well, (**C**) collagen/cell. (**D**–**F**) RDF5, (**D**) nuclei/well, (**E**) collagen/well, (**F**) collagen/cell. Data points are presented as mean values with standard error of the mean error bars. The dotted line on the y axis indicates the mean values obtained for untreated cells. Statistically significant differences were determined using a Brown Forsyth and Welch ANOVA followed by a Dunnett’s T3 multiple comparison test. * *p* < 0.5 and ** *p* < 0.01, compared to the untreated control cells.

**Figure 7 ijms-25-01862-f007:**
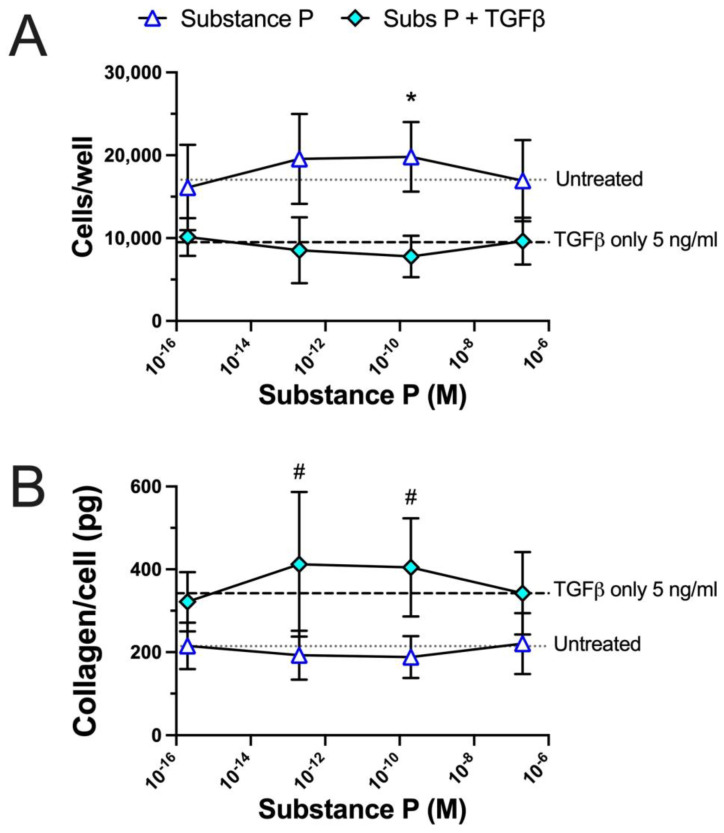
Effect of substance P alone and substance P combined with TGFβ at 5 ng/mL on rat dermal fibroblast proliferation in a 96-well culture system for a total of 6 days in culture. Four concentrations of substance P were used: 2 × 10^--7^ M, 2 × 10^−10^ M, 2 × 10^−13^ M, 2 × 10^−16^ M. The calculated amount of collagen deposited per cell after 6 days of culture is shown. Data from 2 RDF cultures were combined and then normalized to the untreated cells. (**A**) Total nuclear counts in each well. (**B**) Matrix collagen per cell. Data points are presented as mean values with standard deviation error bars. The dotted lines in the figures represent results obtained with untreated cells, while the dashed lines represents result obtained with 5 ng/mL TGFβ-only treated cells. Statistically significant differences were determined using a Brown Forsyth and Welch ANOVA followed by a Dunnett’s T3 multiple comparison test. Significant differences between the substance P-only treatment compared to the untreated cells are designated as * *p* < 0.5. Significant differences between the substance P and TGFβ combined treatment compared to the TGFβ-only treatment are designated as ^#^ *p* < 0.5. See text for other significant findings.

**Figure 8 ijms-25-01862-f008:**
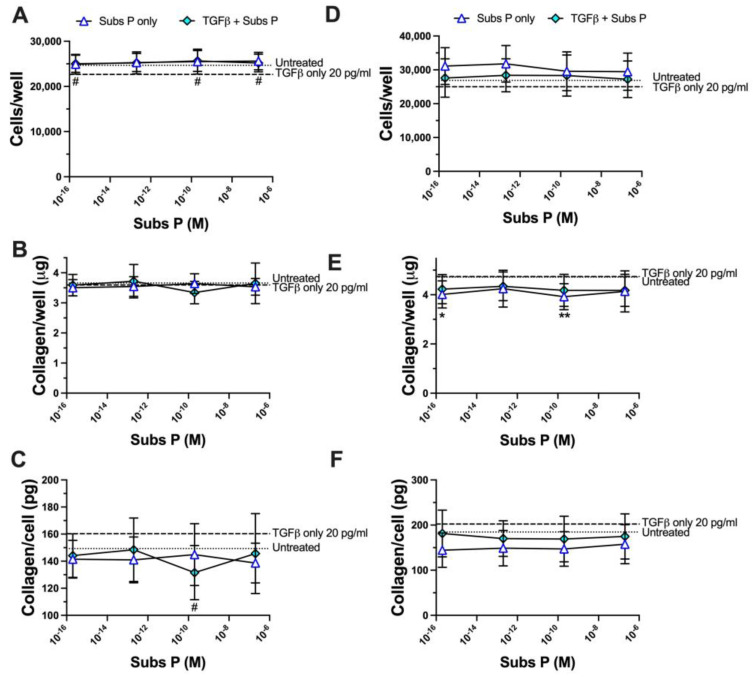
Effect of the combined treatment of TGFβ at 20 pg/mL with varying concentrations of substance P on collagen deposition per cell in a 96-well culture system. Cells were cultured with the treatments for a total of 6 days in 10% FCS containing medium. Concentrations of Substance P used were 2 × 10^−7^ M, 2 × 10^−10^ M, 2 × 10^−13^ M, 2 × 10^−16^ M (**A**–**D**). Combined data from 3 RDF cultures: RDF4, RDF5 and RDF6. (**A**–**C**) Experiment 1, (**D**–**F**) Experiment 2. (**A**,**D**) Number of cells per well, (**B**,**E**) Deposited collagen per well, (**C**,**F**) Deposited collagen per cell. The dotted line in the figure represents results obtained with untreated cells and the dashed line represents the represents results obtained with 20 pg/mL TGFβ only treated cells. Data points are presented as mean values with standard deviation error bars. Statistically significant differences were determined in Graphpad Prism version 10.1.0 using a using a Brown Forsyth and Welch ANOVA followed by a Dunnett’s T3 multiple comparison test. Significant differences between the substance P-only treatment compared to the untreated cells are indicated * *p* < 0.5, ** *p* < 0.01. Significant differences between the substance P and TGFβ combined treatment compared to the TGFβ-only treatment are indicated, ^#^ *p* < 0.5.

**Figure 9 ijms-25-01862-f009:**
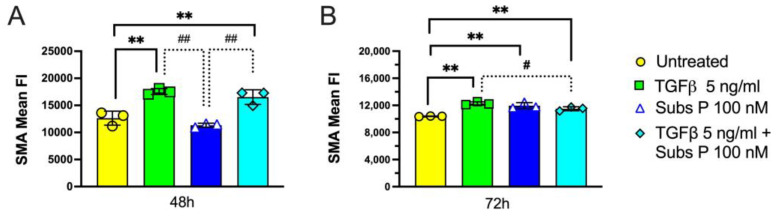
αSMA immunoexpression in rat dermal fibroblasts cultured for 48 h and 72 h in 12-well plates. The results of three independent replicates, per each experimental condition, are shown. Cells were left untreated or were treated with TGFβ 5 ng/mL, substance P 100 nM, or TGFβ (5 ng/mL) and substance P (100 nM) together in a 12-well plate. Cells were trypsinized, transferred to 5 mL tubes, fixed, permeabilized, immunostained for αSMA, and analyzed by flow cytometry. (**A**) 48 h αSMA expression. (**B**) 72 h αSMA expression. Data points are presented as mean values with standard deviation error bars. Statistical significance was determined using a one-way ANOVA followed by a Tukey’s multiple comparison test. ** *p* < 0.01, compared to untreated cells; ^#^ *p* < 0.05 and ^##^ *p* < 0.01, compared between treated groups as shown.

**Figure 10 ijms-25-01862-f010:**
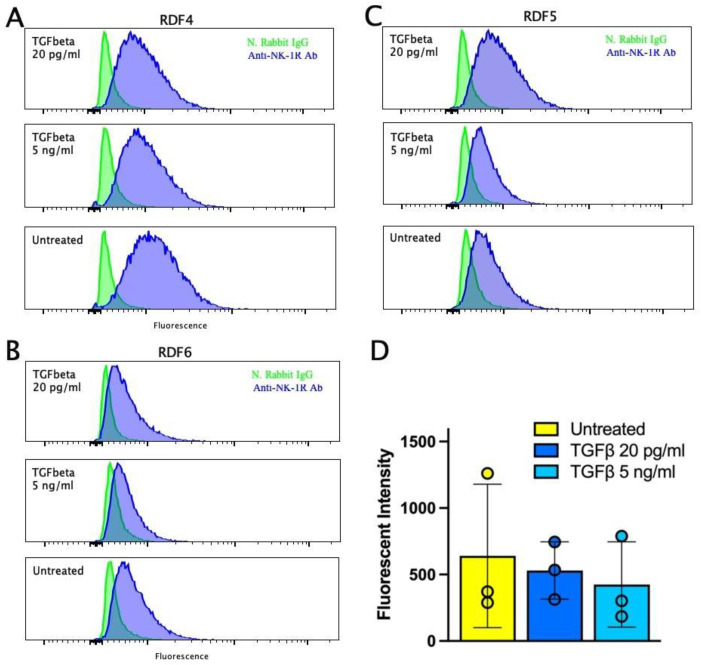
Neurokinin-1 receptor protein expression in RDF cultures. RDFs were cultured in 10 cm dishes and treated with or without TGFβ at 5 ng/mL or 20 pg/mL for two days. Neurokinin-1 receptor was detected using a polyclonal anti-neurokinin 1 antibody. (**A**–**D**) Flow cytometry analysis of anti-NK-1R fluorescence in RDF cultures. Fluorescence histograms shown for cells treated with unlabeled rabbit immunoglobin followed by the anti-rabbit Alexa Fluor 488 secondary antibody or cells stained with the anti-neurokinin-1R antibody followed by the anti-rabbit Alexa Fluor 488 secondary antibody. (**A**) Data obtained with RDF4, (**B**) data obtained with RDF6, (**C**) data obtained with RDF5, (**D**) mean fluorescence intensity quantification of NK-1R expression from the three RDF cultures with standard deviation error bars.

**Table 1 ijms-25-01862-t001:** Statistical outcomes of the numbers of nuclei in day 12 cultures of 4 independent primary cell cultures of rat dermal fibroblasts treated with or without TGFβ or substance P (3-way ANOVA results).

Source of Variation	% of Total Variation	*p* Value	*p* Value Summary		
Cell Cultures	**4.512**	**0.0430**	*****		
Substance P (100 nM)	0.4827	0.3106	ns		
**TGFβ (5 ng/mL)**	**73.23**	**<0.0001**	******		
Cell Cultures × Substance P (100 nM)	2.573	0.1626	ns		
Cell Cultures × TGFβ (5 ng/mL)	3.666	0.0751	ns		
Substance P (100 nM) × TGFβ (5 ng/mL)	**3.165**	**0.0164**	*****		
Cell Cultures x Substance P (100 nM) × TGFβ (5 ng/mL)	**5.329**	**0.0259**	*****		
ANOVA table	SS	DF	MS	F (DFn, DFd)	*p* value
Cell Cultures	33,177	3	11,059	F (3, 16) = 3.416	*p* = 0.0430
Substance P (100 nM)	3549	1	3549	F (1, 16) = 1.096	*p* = 0.3106
TGFβ (5 ng/mL)	538,463	1	538,463	F (1, 16) = 166.3	*p* < 0.0001
Cell Cultures × Substance P (100 nM)	18,918	3	6306	F (3, 16) = 1.948	*p* = 0.1626
Cell Cultures × TGFβ (5 ng/mL)	26,960	3	8987	F (3, 16) = 2.776	*p* = 0.0751
Substance P (100 nM) × TGFβ (5 ng/mL)	23,274	1	23,274	F (1, 16) = 7.189	*p* = 0.0164
Cell Cultures x Substance P (100 nM) × TGFβ (5 ng/mL)	39,183	3	13,061	F (3, 16) = 4.034	*p* = 0.0259
Residual	51,799	16	3237		

Significant findings are bolded; ns = not significant; SS = sum of squares; DF = degrees of freedom; MS = mean of squares. * *p* < 0.05: ** *p* < 0.01.

## Data Availability

Data are contained within the article and in the Appendix A.

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
