# Peer review of "Potentiation of Collagen Deposition by the Combination of Substance P with Transforming Growth Factor Beta in Rat Skin Fibroblasts"

_ijms, 2024, doi:10.3390/ijms25031862_

Round 1

Reviewer 1 Report

Comments and Suggestions for Authors

1. My suggestion, when first writing about SP receptor ligands such as L732,138, the Authors should state whether it is an agonist or antagonist. In the text there is only an abbreviation NK-1RA (line 79), so the reader is not given with the information if it is ago- or antagonist.

2. Fig 1.and Fig. 2 please state in the legend the final concentration of the mixture of SP and TGFbeta for which the results are presented. Similar information should be provided for table 1, especially considering that the Authors examined various doses of such a mixture 

3. The Authors stated that combination of SP and TGFB (Fig.4) resulted in a synergy with regard to collagen deposition. However, in my point of view, it is rather potentiation/additive effect, not synergy

Author Response

Response to 1. “My suggestion, when first writing about SP receptor ligands such as L732,138, the Authors should state whether it is an agonist or antagonist. In the text there is only an abbreviation NK-1RA (line 79), so the reader is not given with the information if it is ago- or antagonist.”

We have indicated that it is an antagonist. Line 88 now reads: Using this model of repetitive injury, we examined whether treatment with a specific human NK-1R antagonist (NK-1RA), L732,138, would prevent the development of fibrosis in musculotendinous tissues [25].

Response to 2. “Fig 1.and Fig. 2 please state in the legend the final concentration of the mixture of SP and TGFbeta for which the results are presented. Similar information should be provided for table 1, especially considering that the Authors examined various doses of such a mixture.” 

We have added these concentrations to the figure legends and to Table 1.

Response to 3. “The Authors stated that combination of SP and TGFB (Fig.4) resulted in a synergy with regard to collagen deposition. However, in my point of view, it is rather potentiation/additive effect, not synergy.”

We changed all wording of synergy to potentiation.

Reviewer 2 Report

Comments and Suggestions for Authors

The paper “Synergistic effects of Substance P with Transforming Growth 2 Factor beta on Collagen Deposition in Rat Skin Fibroblast” is an interesting study and deserves publication. The authors investigated the hypothesis that substance P would increase fibrotic tendencies in rat dermal fibroblasts. They demonstrated that the combined treatment of substance P with TGFb enhances collagen secretion more than either treatment alone in rat dermal fibroblast cultures in a time dependent manner.

The authors investigated the hypothesis that substance P would increase fibrotic tendencies in rat dermal fibroblasts. The authors demonstrated that the combined treatment of substance P with TGFb enhances collagen secretion more than either treatment alone in rat dermal fibroblast cultures in a time dependent manner. The obtained results differ from previous results showing that tenocytes secreted more collagen I after substance P treatment (with or without TGFb co-treatment), compared to TGFb alone, at 24 hr of culture; yet similar amounts of collagen secretion after treatment with substance P, TGFb, or substance P and TGFb co-treatment by 48 hr of culture. Only TGFb increased smooth muscle actin at 48 hr of culture.

The demonstrated synergy was dependent on the concentration of each factor. May be in future to check more this dependence. The obtained data suggest that the effects of substance  P on rat dermal fibroblasts can be mediated through ligation of the NK-1R expressed on the cells. However, substance P effects can also be mediated through the other neurokinin receptors NK-2R and NK-3R. The authors should check this in future experiments.
The conclusions are consistent with the evidence and arguments presented.
The references are appropriate.
It is better to use the same units for concentration, nM: “TGFb (5 ng/ml) and substance P (100 nM)”.

Author Response

Author's Reply to the Review Report (Reviewer 2)

Response to: “The paper “Synergistic effects of Substance P with Transforming Growth 2 Factor beta on Collagen Deposition in Rat Skin Fibroblast” is an interesting study and deserves publication. The authors investigated the hypothesis that substance P would increase fibrotic tendencies in rat dermal fibroblasts. They demonstrated that the combined treatment of substance P with TGFb enhances collagen secretion more than either treatment alone in rat dermal fibroblast cultures in a time dependent manner.”

Thank you.

Response to: “The authors investigated the hypothesis that substance P would increase fibrotic tendencies in rat dermal fibroblasts. The authors demonstrated that the combined treatment of substance P with TGFb enhances collagen secretion more than either treatment alone in rat dermal fibroblast cultures in a time dependent manner. The obtained results differ from previous results showing that tenocytes secreted more collagen I after substance P treatment (with or without TGFb co-treatment), compared to TGFb alone, at 24 hr of culture; yet similar amounts of collagen secretion after treatment with substance P, TGFb, or substance P and TGFb co-treatment by 48 hr of culture. Only TGFb increased smooth muscle actin at 48 hr of culture.”

Correct. Thank you.

Response to: “The demonstrated synergy was dependent on the concentration of each factor. May be in future to check more this dependence. The obtained data suggest that the effects of substance  P on rat dermal fibroblasts can be mediated through ligation of the NK-1R expressed on the cells. However, substance P effects can also be mediated through the other neurokinin receptors NK-2R and NK-3R. The authors should check this in future experiments.”

Done: Lines 565-9 – now read: “We tested two different concentrations of TGFb with several concentrations of substance P to demonstrate the positive potentiation of collagen deposition. Considering the proliferative effect of lower concentrations of TGFb, other concentrations of TGFb, especially lower concentrations, could be tested to further investigate the interaction of the two proteins.

Lines 585-6 now read: “However substance P effects can also be mediated through the other neurokinin receptors NK-2R and NK-3R. Future experiments to address these possibilities are planned.”

The conclusions are consistent with the evidence and arguments presented.
The references are appropriate.

Response to: “It is better to use the same units for concentration, nM: “TGFb (5 ng/ml) and substance P (100 nM)”.

We added an explanation on lines 128-131:” (We use ng/ml instead of nanomolar in reference to TGFb concentration for reasons of comparison with our previous publications and many other publications available in the literature; with TGFb having a molecular weight of 25 kiloDaltons, 5ng/ml is equal to 200pM of TGFb).”